# Assessment of a New Solar Radiation Nowcasting Method Based on FY-4A Satellite Imagery, the McClear Model and SHapley Additive exPlanations (SHAP)

**Dongyu Jia [1,\*], Liwei Yang [2], Xiaoqing Gao [2] and Kaiming Li [1]**

[1] College of Urban Environment, Lanzhou City University, Lanzhou 730070, China
[2] Key Laboratory of Land Surface Process and Climate Change in Cold and Arid Regions/Northwest Institute of Eco-Environment and Resources, Chinese Academy of Sciences, Lanzhou 730000, China
\* Correspondence: jiadongyu@lzcu.edu.cn

**Abstract:** The global warming effect has been accelerating rapidly and poses a threat to human survival and health. The top priority to solve this problem is to provide reliable renewable energy. To achieve this goal, it is important to provide fast and accurate solar radiation predictions based on limited observation data. In this study, a fast and accurate solar radiation nowcasting method is proposed by combining FY-4A satellite data and the McClear clear sky model under the condition of only radiation observation. The results show that the random forest (RF) performed better than the support vector regression (SVR) model and the reference model (Clim-Pers), with the smallest normalized root mean square error (nRMSE) values (between 13.90% and 33.80%), smallest normalized mean absolute error (nMAE) values (between 7.50% and 24.77%), smallest normalized mean bias error (nMBE) values (between −1.17% and 0.7%) and highest $R^2$ values (between 0.76 and 0.95) under different time horizons. In addition, it can be summarized that remote sensing data can significantly improve the radiation forecasting performance and can effectively guarantee the stability of radiation predictions when the time horizon exceeds 60 min. Furthermore, to obtain the optimal operation efficiency, the prediction results were interpreted by introducing the latest SHapley Additive exPlanation (SHAP) method. From the interpretation results, we selected the three key channels of an FY-4A and then made the model lightweight. Compared with the original input model, the new one predicted the results more rapidly. For instance, the lightweight parameter input model needed only 0.3084 s (compared to 0.5591 s for full parameter input) per single data point on average for the 10 min global solar radiation forecast in Yuzhong. Meanwhile, the prediction effect also remained stable and reliable. Overall, the new method showed its advantages in radiation prediction under the condition that only solar radiation observations were available. This is very important for radiation prediction in cities with scarce meteorological observation, and it can provide a reference for the location planning of photovoltaic power stations.

**Keywords:** global solar radiation; forecast; machine learning models; SHapley Additive exPlanation





## 1. Introduction

In recent decades, with the continuous growth of the global population and excessive carbon emissions, the global greenhouse effect have been accelerating rapidly. These factors pose a threat to human survival and health [1]. The United Nations Environment Program claimed that the emission reduction target of each country is far from the 1.5 °C target [2]. More and more countries are reducing the emissions gap by participating in climate actions, such as carbon neutrality. To achieve the goal of carbon neutrality and meet industrial and living requirements in the foreseeable future, it is necessary to replace traditional energy with renewable energy. Because of its outstanding performance in noise, carbon emissions and operations, solar energy has an obvious advantage over other renewable energies [3].

As one of the largest carbon emitters, most of China's electricity is still supplied by fossil fuels. However, the widespread use of traditional energy emits large amounts of greenhouse gases (such as $CH_4$, $SO_2$ and CO), which is not conducive to meeting China's commitment to achieve carbon neutrality by 2060 [1,4]. Therefore, the extensive use of solar energy is one of the important ways to achieve this goal. However, contrary to traditional power, solar electricity generation is impacted by transient clouds, aerosols and weather and is, thus, highly intermittent. This affects the layout and development of the photovoltaic industry [5–7]. The maintenance cost of PV can be controlled and predicted through the prior knowledge of the expected power output. Accurate predictions of solar radiation are the premise for realizing photovoltaic power generation, and they are one of the most important bases for establishing photovoltaic electric fields.

By summarizing previous studies, solar forecasting methods can be simply classified into four groups by data type and method [8]: (1) data-driven methods (e.g., [9,10]), (2) camera-based methods (e.g., [11,12]), (3) satellite-based methods (e.g., [13,14]) and (4) numerical weather predictions (NWP) (e.g., [15]). The data-driven method is applicable to all forecast horizons, but it is usually applied to 1-h ahead forecasting. While camera-based and satellite-based methods generally forecast based on continuous images, the forecasting time of the camera-based method is usually within 1 h, and the forecasting time of the satellite-based method can be extended from 0.5 h to 6 h. In terms of estimation algorithms, the existing universally applicable methods are relatively high on a large scale, but their calculation speed is greatly reduced when the number of input parameters is large. Consequently, current methods cannot easily meet the needs of real-time observation and cannot cope with massive satellite data. In addition, under the premise of ensuring inversion accuracy, attention should be given to the efficiency of the data and the improvement of inversion efficiency in the future, such as through the efficient combination of look-up table (LUT) methods and deep learning and other artificial intelligence methods to promote the development of high-precision solar radiation products [11–14]. At the same time, the data-driven prediction method requires multiple data inputs from a first-class (meteorological) station, which are rarely available due to potential calibration problems [16]. Therefore, how to provide solar forecasts with high spatial and temporal resolutions at a low cost has become an essential research topic. For radiation forecasts, the choice of research methods is also crucial. Most of the existing methods cannot solve the complex nonlinear relationships in a noisy environment [17]. Therefore, an increasing number of researchers have chosen to apply machine learning methods to solar radiation prediction with the development of artificial intelligence. The research shows that the artificial neural network can complete radiation prediction tasks well. Deo et al. [18] developed a support vector machine-wavelet-coupled model (WSVM). By inputting meteorological observation parameters, the model can output reliable daily forecast horizons. Yagli et al. [19] combined this system with a machine learning method to correct the bias. The research results show that the bias-corrected satellite-derived data can generate prediction products with the same prediction accuracy as the observation data.

The above research proves the feasibility of using machine learning in radiation prediction. However, most of the regions in China where large-scale PV power generation is planned lack the support of meteorological or other observation data. Therefore, how to use the limited radiation observation data to obtain high spatial and temporal resolution radiation predictions is worthy of in-depth study. By analyzing the main solar radiation-influencing factors, this study selected McClear (a clear sky model) and FY-4A data, which are easy to obtain, and developed a fast and accurate solar radiation nowcasting method based on machine learning. Based on the SHAP interpretation results, we selected three key channels on the FY-4A to improve the calculation speed of the model, and the role of remote sensing data in radiation prediction is also discussed. The new method can reduce the dependence of radiation predictions on other meteorological observation data and can be used to optimize the selection of geographical locations for photovoltaic power stations.

## 2. Data and Methodology

### 2.1. Introduction of the Research Process

In this study, FY-4A satellite imagery, solar radiation observations and a McClear-based machine learning regression model were developed to obtain the accurate global solar radiation forecasting. We compared two machine learning models that predict global solar radiation in Yuzhong, Minqin and Dunhuang. The datasets of the three urban indicators related to solar radiation were collected from July 2019 to June 2020. The operation steps of the method are briefly shown in in Figure 1. First, the data underwent quality control and a pretreatment. The radiation observation data at the current time, remote sensing data (different band combinations) and McClear data were used as the input parameters for machine learning, and the radiation observation data corresponding to the predicted time step were used as the output parameters. Data were randomly grouped into training and testing datasets at proportions of 70% and 30%, respectively. Next, the RF and SVR algorithms were introduced for model building. In order to find the optimal parameters, the grid search method was adopted to derive the hyperparameters. Furthermore, performance evaluations were performed based on the statistical indicators. Related indicators, such as nMAE, nMBE, nRMSE, Skillscore and R, were utilized to quantify the evaluation results. Then the optimal machine learning method was selected based on the comprehensive performance. Finally, in the interpretation and lightweight process, the lightweight parameter input model (based on the SHAP results) was established, and a comparison with the initial model in the same field was also made.

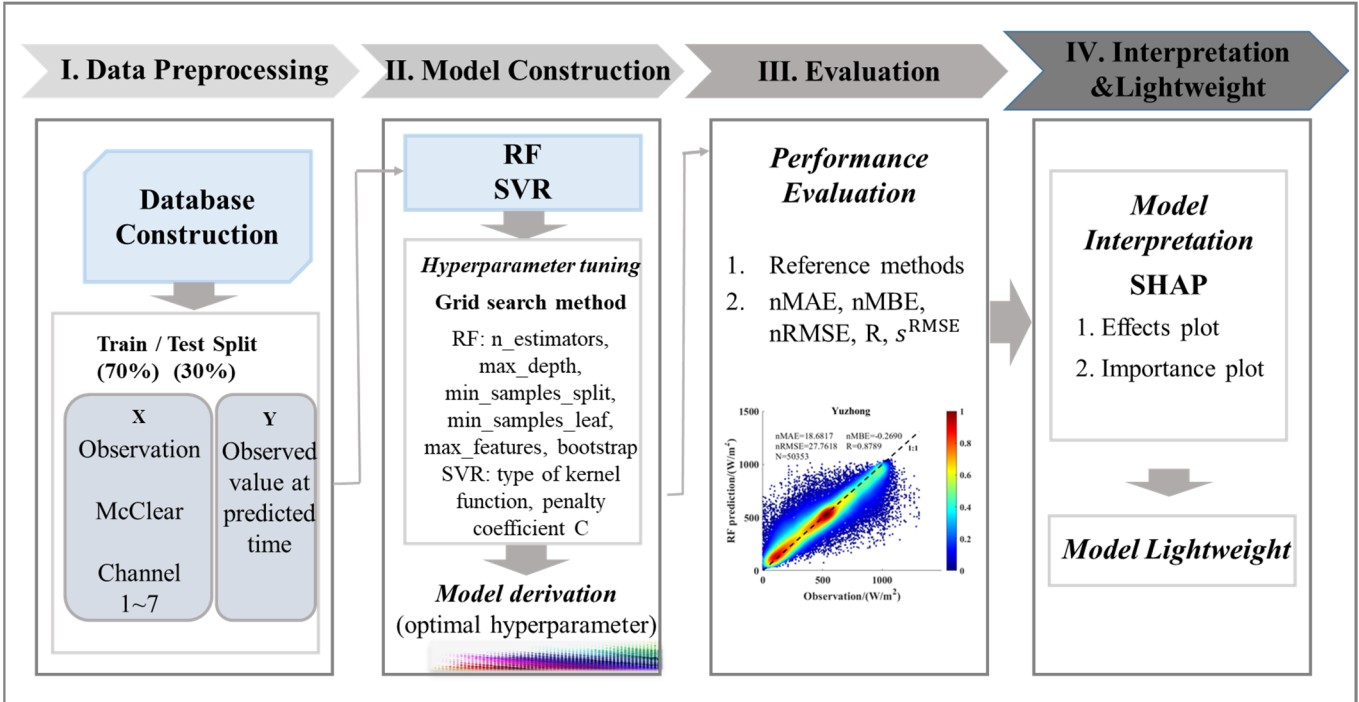

**Figure 1.** Research flowchart.

### 2.2. Data

#### 2.2.1. FY-4A Satellite

The FY-4A used the SAST5000 platform and had a hexagonal cylinder structure, which had the advantages of a large surface area and a low centroid. The FY-4A satellite was equipped with advanced observation instruments, including an advanced interferometric atmospheric vertical detector, a space environment-monitoring instrument, a geostationary orbit radiation imager and a lightning imager.

The cloud image data of the Fengyun-4A satellite came from the National Meteorological Science Data Center (http://satellite.nsmc.org.cn/, accessed on 1 December 2016).

The FY-4A carried a variety of payloads, including an advanced geosynchronous radiation imager (AGRI) (14 channels between 0.47 and 13.5 μm, with a spatial resolution of 0.5–4 km), an interferometric atmospheric vertical detector (GIIRS), a lightning imager (LMI) and a space environment-monitoring instrument (SEP). The spatial resolution of satellite images used in this study was 2 km × 2 km, and seven channels (0.45~4.00 μm; three channels for visible light, three channels for short-wave infrared and one channel for medium-wave infrared) were selected. First, we deleted the satellite images whose solar altitude angle was less than 10 degrees (the observation error was large due to the weak light). Secondly, geometric calibration and radiometric calibration were carried out for the cloud image. The cropped area of the cloud image was 32 km × 32 km, and the pixels in the inversion area were in the middle. Finally, the missing cloud images were interpolated using linear interpolation, and the regional average of the albedo of the 7 channels with a 10 min time resolution (matching the resolution of observation data) was obtained.

### 2.2.2. McClear Data

The McClear is a fully physical model parameterized by A, z and several parameters describing the optical state of the atmosphere. For clear-sky conditions, an irradiation time series is provided for any location in the world using information on aerosol, ozone and water vapor from the CAMS global forecasting system. Other properties, such as the ground albedo and ground elevation, are also considered. Similar time series are available for cloudy conditions, but since the high-resolution cloud information is directly inferred from satellite observations, these are currently only available inside the field-of-view of the Meteosat Second Generation (MSG) satellite, which is roughly over Europe, Africa, the Atlantic Ocean and the Middle East. Input quality control, regular quarterly benchmarking against ground stations and regular monitoring of the consistency and detecting of possible trends are performed.

In recent years, many studies have proved the reliability of McClear (including in China) [20,21]. Among them, global, direct and diffuse horizontal irradiations, as well as the beam normal irradiations of 1 min, 15 min, 1 h, daily and monthly time scales can be obtained directly by inputting longitude, latitude, altitude and output format (https://www.soda-pro.com/web-services/radiation/cams-mcclear, accessed on 1 January 2004). The global solar radiation of Yuzhong, Minqin and Dunhuang from June 2019 to July 2020 were downloaded with 1 min temporal resolution as one of the machine learning input parameters.

### 2.2.3. Observation

Ground solar radiation data were obtained from three meteorological observation stations in the Northwestern China, namely Yuzhong, Minqin and Dunhuang (Figure 2). Northwestern China accounts for 60% of the total installed capacity and grid of connected power in China due to its simple vegetation types, sparse population and arid climate. According to the standards of ISO 9060:1990 and WMO, all instruments were calibrated and maintained in time. The dataset used was from July 2019 to June 2020, and the temporal resolution was 1 min. These stations provided data of short-wave downward global radiations in W/m$^2$, which were measured using the FS-S6 solar radiation sensor. The observation data in the input data were real-time matched with the remote sensing data, while the radiation data in output data were delayed matched based on the predicted time step. The geographic statistics of the 3 training and testing study sites are also listed in Table 1.

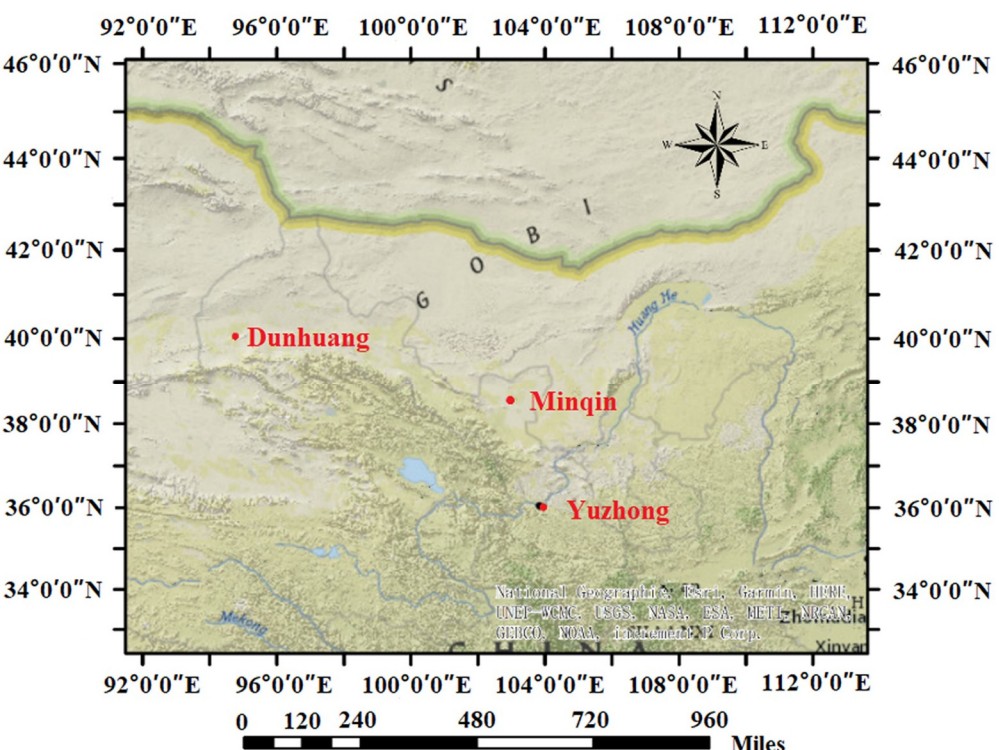

**Figure 2.** Spatial distribution of the radiation observation sites (red marks).

**Table 1.** Geographic statistics of the 3 training and testing study sites.

| Station Name | Latitude (°) | Longitude (°) | Elevation (m) | Mean GHI (W/m²) | Max GHI (W/m²) | Data Size |
|---|---|---|---|---|---|---|
| Yuzhong | 35.87 | 104.15 | 1874.4 | 460.36 | 1307 | 19,525 |
| Minqin | 38.63 | 103.09 | 1367.5 | 520.22 | 1289 | 18,840 |
| Dunhuang | 40.15 | 94.68 | 1139.0 | 548.17 | 1219 | 18,024 |

GHI: Global Horizontal Irradiance.

*2.3. Methods*

2.3.1. Machine Learning

According to previous research [3], SVR and RF are the best two models for estimating and predicting solar radiation in China's regions. Therefore, this study used the following two methods to construct radiation prediction models and then selected the optimal model from them.

1.　SVR model

The SVR model can be simply understood as creating a "gap zone" on both sides of the linear function, and the distance of this "gap zone" is $\epsilon$ (this value is often given according to experience). The optimized model is obtained by minimizing the total loss and maximizing the gap. Nonlinear models such as SVR can be spatially mapped by using kernel functions, and then the predicted values can be obtained by regression [22].

The prediction of $x_*$ by the SVR method is given in the following formula:

$$\hat{y} = \sum_{i=1}^{n} \alpha_i k_{rbf}(x_i, x_*) + b \tag{1}$$

Parameter $b$ can be calculated from specific conditions, and $\alpha_i$ can be calculated by the difference between two Lagrange multipliers.

2.　RF model

The RF (random forest) method is an improved bagging regression tree. Because of its wide range of applications, high precision, difficulty in over-fitting and ability

to process nonlinear data, the random forest method has been widely used in predictions and classifications in different fields [23]. RF theory was introduced in detail in Yu's research [24].

Super-parameters were set in advance before training. In the study, the grid search was selected to optimize the super-parameters. For the SVR method, the hyperparametric optimization was mainly aimed at the type of kernel function (radial basis function, RBF, or linear kernel) and the penalty coefficient C. For RF, the hyperparametric optimization was mainly aimed at the following six super-parameters: the number of decision trees (n_estimators), the maximum depth of decision trees (max_depth), the minimum number of separated samples (min_samples_split), the minimum number of leaf node samples (min_samples_leaf), the maximum number of separated features (max_features) and whether to conduct random sampling (bootstrap).

3. Reference model (Clim-Pers, combination of climatology and persistence model)

In order to ensure the comparability of the prediction models, this study introduced the reference method proposed by Yang for comparison. Yang effectively combines classic climatology and persistence methods by calculating lag autocorrelation [25]. The core parameters of the reference method are as follows.

The combination of climatology and the persistence model is chosen as the reference forecasts, namely,

$$y_i = \alpha x_{i-h} + (1-\alpha)\mu \tag{2}$$

where $\alpha$ represents the weight on the persistence model, and $1 - \alpha$ is the weight on the climatology model and $y_i$ denotes the forecast value, respectively. For a variable of interest, $x$, the *RMSE* of the internal single-valued climatology is given by $RMSE_c$. The *RMSE* of the climatology-persistence combination is thus given by $RMSE_{cp}$.

$$RMSE_c = \sqrt{\frac{1}{n}\sum_{i=1}^{n}(\mu - x_i)^2} = \sigma \tag{3}$$

$$\begin{aligned} RMSE_{cp} &= \sqrt{\frac{1}{n}\sum_{i=1}^{n}[\alpha x_{i-h} + (1-\alpha)\mu - x_i]^2} \\ &= \sqrt{\alpha^2\sigma^2 + \sigma^2 - 2\alpha\text{cov}(x_{i-h}, x_i)} \end{aligned} \tag{4}$$

$$\alpha = \frac{\text{cov}(x_{i-h}, x_i)}{\sigma^2} = \gamma(h) \tag{5}$$

where $x_{i-h}$ denotes the lag-h autocorrelation (assuming the end effect is negligible), $\mu$ and $\sigma$ denote the sample mean and standard deviation (assuming n is large).

### 2.3.2. Performance Evaluation

In order to evaluate the performance of the models, the Pearson correlation coefficient ($R$), the normalized mean absolute error ($nMAE$), the normalized root mean square error ($nRMSE$) and the normalized mean bias error ($nMBE$) were selected, as follows:

$$MAE = \frac{1}{N} \times \sum_{i=1}^{N}|y_i - x_i| \quad nMAE = \frac{MAE}{\overline{x}} \times 100\% \tag{6}$$

$$MBE = \frac{1}{N} \times \sum_{i=1}^{N}(y_i - x_i) \quad nMBE = \frac{MBE}{\overline{x}} \times 100\% \tag{7}$$

$$RMSE = \sqrt{\frac{1}{N} \times \sum_{i=1}^{N}(y_i - x_i)^2} \quad nRMSE = \frac{RMSE}{\overline{x}} \times 100\% \tag{8}$$

$$R = \frac{\sum\limits_{i=1}^{N} (x_i - \overline{x})(y_i - \overline{y})}{\sqrt{\sum\limits_{i=1}^{N} (x_i - \overline{x})}\sqrt{\sum\limits_{i=1}^{N} (y_i - \overline{y})}} \tag{9}$$

where $x_i$ and $y_i$ represent the measurements and predictions, respectively.

Although *RMSEs* are commonly used in radiation forecasting, in order to better reflect the accuracy of the prediction model, the continuous ranked probability score (skill score) was chosen in the probabilistic solar forecasting [26]:

$$s^{\mathrm{RMSE}} = 1 - \frac{\mathrm{RMSE}_{model}}{\mathrm{RMSE}_{cp}} \tag{10}$$

## 3. Results and Discussion

To avoid the impact of sample differences during the machine learning model training and testing processes, 70% of the data were randomly selected as the training data (the remaining 30% were selected as testing data) by using the random module in Python. These randomized data included the radiation observation data at the current time series, the McClear data and the regional average of cloud image albedos in different channels. It can be seen in Figure 3 that when the total amount of learning data was limited (the total amount of data used in this manuscript was from 1 year), the advantage of using random functions to select learning arrays was that they could contain various change rules to the greatest extent.

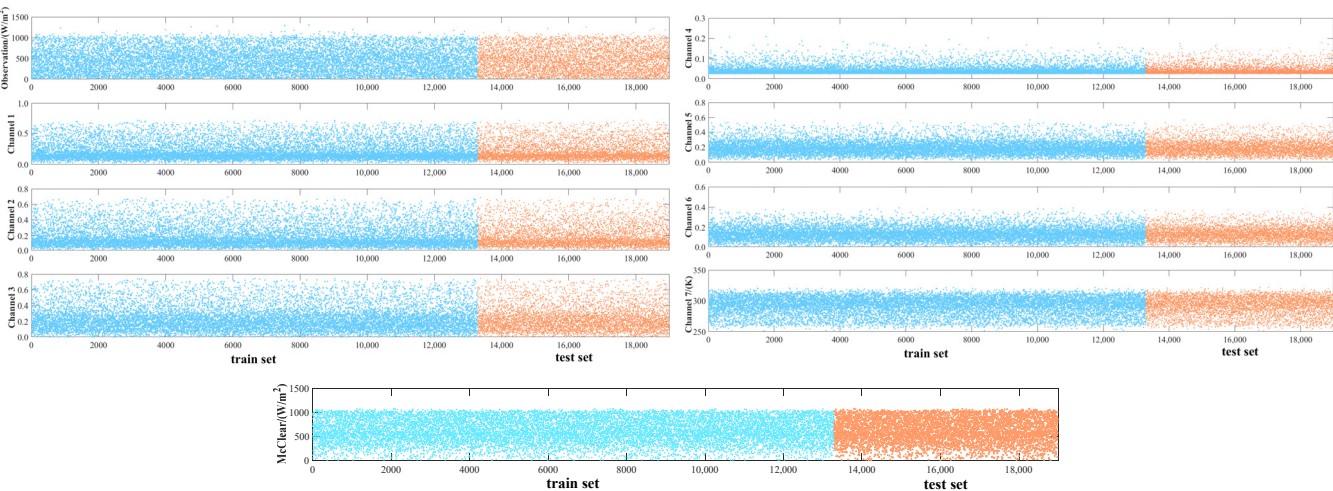

**Figure 3.** Random distribution of machine learning input and testing parameters (for instance, location: Yuzhong, time horizon: 10 min, train set (N) = 13,290, test set (N) = 5696).

### 3.1. Assessing the Applicability of Machine Learning Methods

In order to accurately evaluate the solar radiation forecasting performances of different machine learning models, according to previous research [18], the two most effective machine learning models were selected (RF and SVR). At the same time, the effect of the outputting test data was also more reliable. The performance characteristics of the models were evaluated by utilizing the nRMSE, nMAE, nMBE and R.

The scatter diagrams of the measured and predicted values (RF and SVR) are presented for Yuzhong, Minqin and Dunhuang (Figure 4). The red and dark blue dots represent the high-density and low-density samples, respectively. This figure also presents the values of nMAE, nMBE, nRMSE, N (the number of samples) and the correlation coefficient (R), in addition to the 1:1 line (y = x).

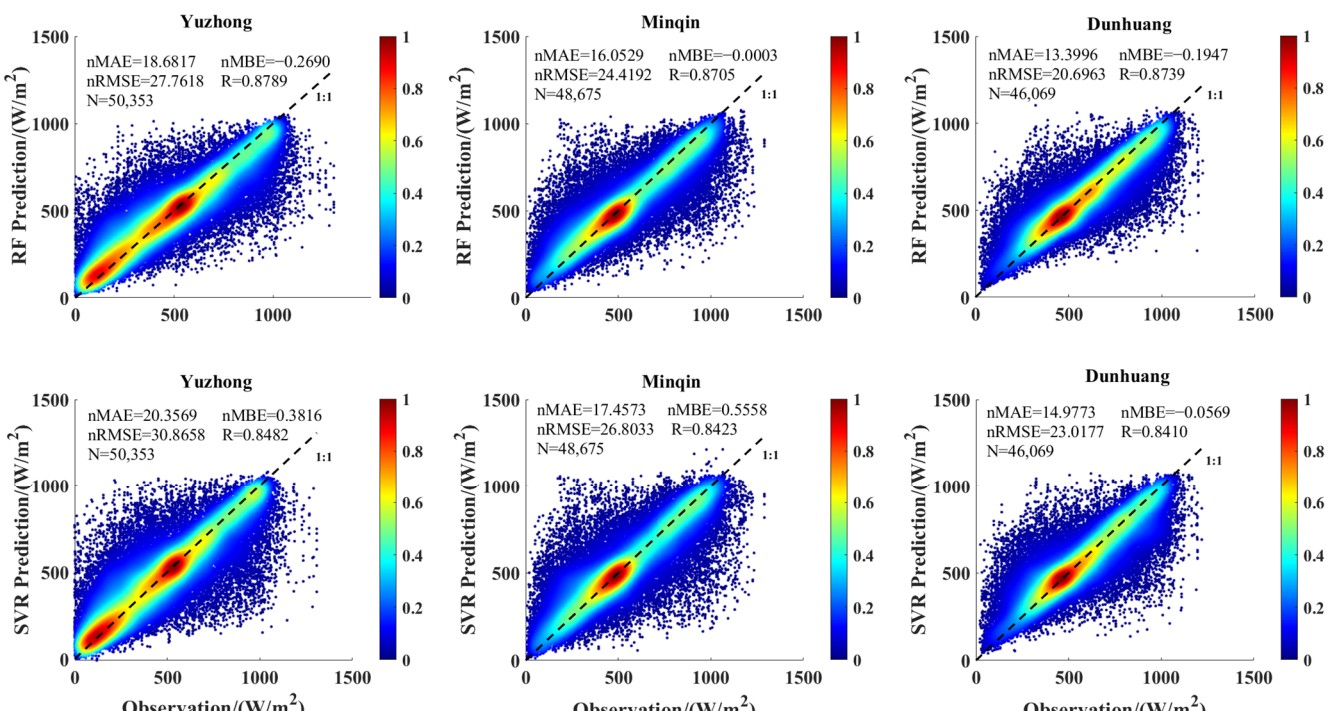

**Figure 4.** Scatter diagrams of the measurements (horizontal axis) and forecasts (vertical axis) of the GHI for Yuzhong, Minqin and Dunhuang. (The first row and the second row represent the overall forecast performance of RF and SVR, respectively, on the GHI with different prediction durations.) The color represents the frequency of each pair: R, the Pearson correlation coefficient, and RMSE, the root mean square error.

The results showed that the overall RF and SVR prediction effects in each region were similar. In all study areas, the global solar radiation predictions utilizing RF were better than those utilizing SVR. According to the nMBE values in the figure, it can be clearly seen that using SVRs to predict the values may have a greater probability of positive deviations. Among them, the normalized mean bias error (nMBE) values of the global solar radiation predicted by the RF method in Minqin were the smallest (−0.0003%). The normalized root mean square error (nRMSE) values of global solar radiation utilizing RF in the Dunhuang area were the smallest (20.6963%). At the same time, the Pearson correlation coefficient (R) values were also the largest (0.8739) in Dunhuang (utilizing RF).

### 3.2. Comparison of the Accuracy Achieved by the RF, SVR and Reference Models

To analyze the rule of the forecast effect with the time horizons, the forecasting performances of 10, 20, 30, 40, 50, 60, 90, 120, 150 and 180 min are shown in Figure 5. The nRMSE, nMAE, nMBE and $R^2$ metrics for the three methods are also shown in Figure 5.

The RF forecasts had a time-horizon averaged nRMSE of 16.36% at 10 min, and it increased steadily with the forecast time to 30.36% at 180 min. The nMAE values had a similar trend, which was 7.50% at 10 min and increased to 21.98% at 180 min (time horizon averaged). The R value was 0.95 at 10 min and decreased rapidly to 0.78 at 180 min (time horizon averaged). Unlike the other metrics, the values of nMBE fluctuated around 0. The nRMSE improvements of RF (and SVR) compared to Clim-Pers were 16.35 (15.00) percent points at 10 min and 35.84 (22.54) percent points at 180 min. Compared with the Clim-Pers reference model, the longer the time horizon was, the more obvious the improvement of the prediction effect.

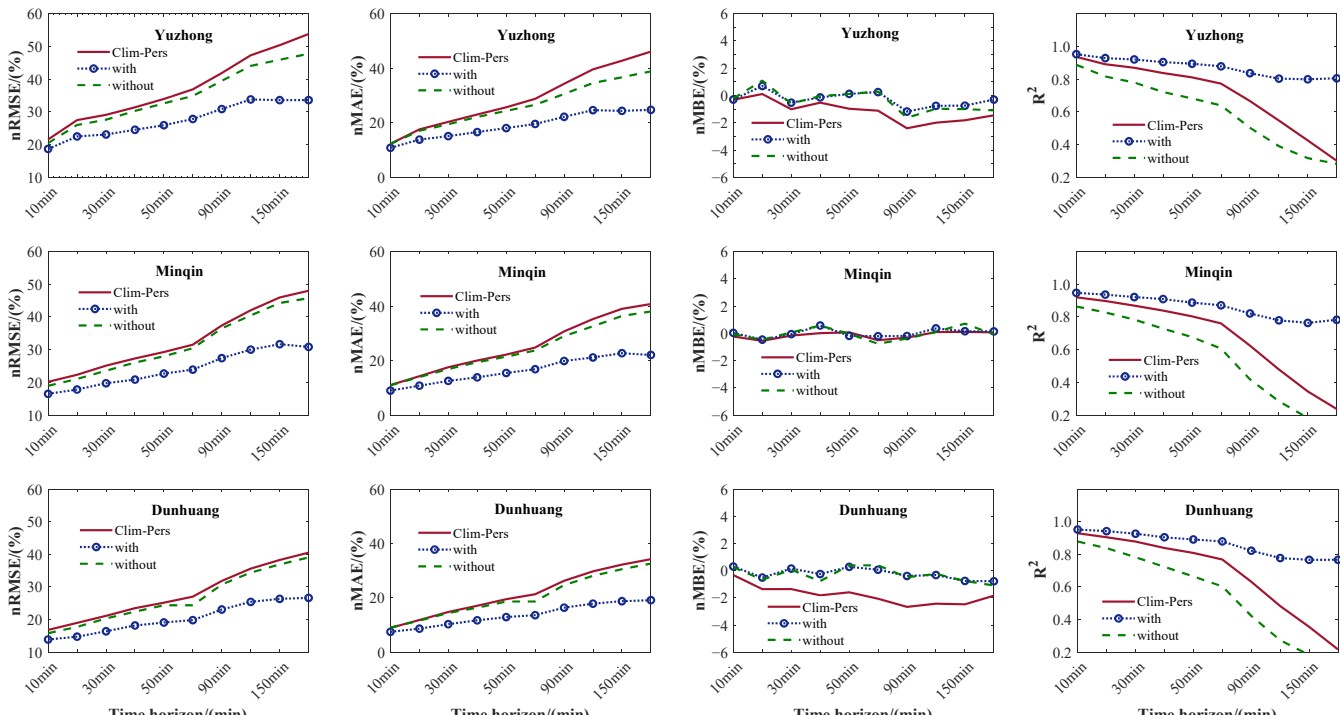

**Figure 5.** Site-specific nRMSE, nMAE, nMBE and $R^2$ values by forecast time for the three compared methods.

The small nMAE and nMBE values of the Clim-Pers method are understandable, as the starting point was the observed irradiance. Unlike the forecast results, the Clim-Pers method underestimated the value of most in situ irradiance for all forecast times (based on the nMBE), especially in Dunhunag. The Clim-Pers had larger nRMSE and nMAE values than RF and SVR at all sites, but the $R^2$ values of the Clim-Pers results were smaller. The main difference between Clim-Pers, RF and SVR may be due to cloud information, because remote cloud field development-sensing information was successfully added to RF and SVR. For the Clim-Pers model, the better accuracy at the start of the forecast was rapidly lost, resulting in higher nRMSE values after 60 min of forecast time at each site.

In general, compared with the reference model (Clim-Pers) and SVR, RF performed the best, with the time horizon averaging the smallest nRMSE values (between 13.90% and 33.80%), smallest nMAE values (between 7.50% and 24.77%), smallest nMBE values (between −1.17% and 0.7%) and highest $R^2$ values (between 0.76 and 0.95).

Based on the reference model, we analyzed two machine learning methods in depth. A skill score comparison for Yuzhong, Minqin and Dunhuang is shown in Figure 6. In this case, RF was found to perform better at all three sites, especially when the time horizon was longer than 30 min. In the solar radiation forecasts of the three sites, the difference range of the SVR and RF skill scores changed from 0.0025–0.016 (0–30 min) to 0.012–0.140 (40–180 min). Among them, the maximum values of the skill score differences of the three sites all occurred at 180 min, with a range of 0.12–0.14.

In summary, there are few differences between the two radiation prediction machine learning models within 30 min, but RF showed better stability with increasing time horizons. Therefore, RF is a better machine learning method for building a solar radiation forecast model based on global solar radiation observations using McClear and satellite data in northwestern China. Compared with the research results of Ravinesh et al. [27], both methods can provide radiation predictions within an acceptable range. However, our method can provide more stable radiation prediction results as the time step increases, and it requires fewer operational and input parameters.

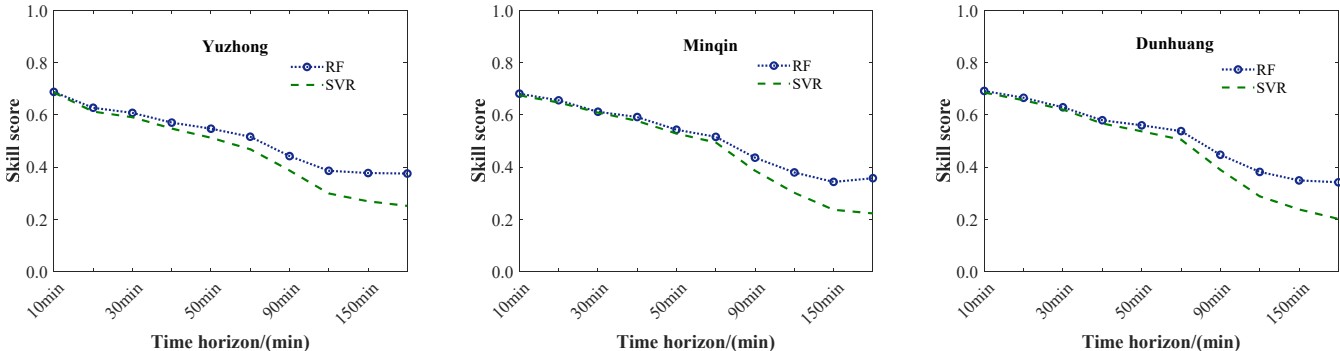

**Figure 6.** RF and SVR forecast skills by forecast time against the Clim-Pers method.

### 3.3. Importance Analysis and Lightweight Model Discussion

This research shows that the model's running time is closely related to the number of input variables. A long running time and high-level computing requirements are not conducive to the future deployment of the model in the photovoltaic industry. Therefore, under the condition of maintaining the prediction accuracy, selecting input parameters that are highly related to the prediction results becomes increasingly important. To analyze the influence of the different input variables on the radiation prediction results at different time horizons in RF, the data of 10, 60, 120 and 180 min were interpreted by SHAP (SHapley Additive exPlanations).

The SHAP values of each feature are shown in the SHAP summary plot (Figure 7). The relationship between the size of the feature values and the predicted impacts can be seen through the color, and the distributions of the feature values are also displayed. On the *X*-axis, the output of the model is listed. On the *Y*-axis, features sorted according to their contributions are represented. This can clearly show the contribution of each feature to the overall forecasting. Therefore, the larger the average SHAP value is, the more important the feature is. Based on the SHAP results, the top five input variables with the highest impact on the model output were Observation, McClear, Channel 7, Channel 2 and Channel 1. In addition, Observation made the most important contribution to the model output, especially for ultrashort-term radiation forecasts within 60 min. The observation and forecast data show an obvious positive correlation, regardless of the time horizon. However, Channels 1 and 2 showed opposite correlations with the forecasts, and Channel 7 and McClear showed a positive correlation with the radiation forecasts within 60 min. It is worth noting that the correlation is not obvious over the range of 90 to 180 min.

In summary, among the input parameters, Observation, McClear and Channels 1, 2 and 7 had greater contributions to the global solar radiation forecast results. The impact of adding remote sensing data on the effectiveness of radiation predictions will be discussed in detail in the next section.

Table 2 shows that adding satellite data to the input data can significantly improve the radiation predictions. It is worth noting that the prediction effect without adding remote sensing data decreased rapidly after the time horizon exceeded 60 min. The specific details are that the values of the nRMSE and the nMAE continued to increase, and the prediction accuracy decreased rapidly. Among them, the range of nRMSE differences (excluding remote sensing data minus including remote sensing data) changed from 3.8777~6.044% with a time horizon of 60 min to 10.0181~13.8113% with a time horizon of 180 min; $R^2$ changed from 0.6166 with a time horizon of 60 min to 0.1795 with a time horizon of 180 min. Therefore, it can be concluded that remote sensing data can improve radiation predictions and effectively guarantee the stability of radiation predictions when the time horizon exceeds 60 min.

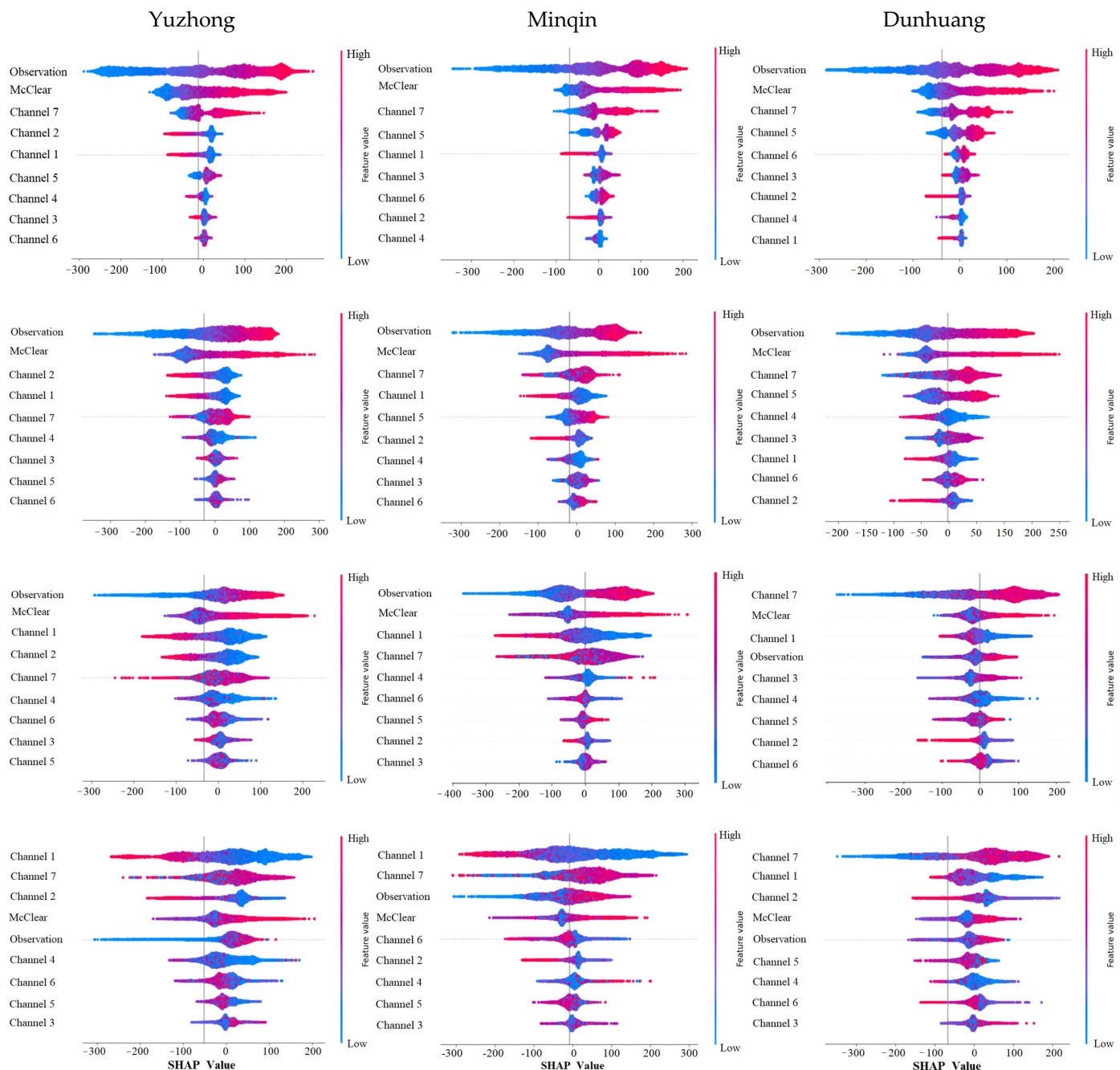

**Figure 7.** SHAP summary plots of the three observation sites in typical time horizons (red is the high value and blue is the low value).

In addition, the selection of remote sensing data channels can improve the operational efficiency of radiation predictions. By comparing the relevant data presented in Table 2, the variation ranges of the model running time differences between the full-channel and the three-channel models were from 585.9727 s to 1025.7605 s. At the same time, it is notable that the lightweight parameter input model had similar relevant performances to the all-parameters input model at any time horizon and that they had most of the same $R^2$, nMAE, nRMSE and nMBE values.

**Table 2.** Comparison of the mean radiation prediction effect under different time horizons.

| Time Horizon | Input Data (McClear and Radiation Observations Are Not Listed) | nRMSE/% | nMAE/% | nMBE/% | $R^2$ | T/s |
|---|---|---|---|---|---|---|
| 10 min | FY-4A (Best 3 Channels) | 16.5793 | 9.3844 | 0.0079 | 0.9011 | 2328.9037 |
| | FY-4A (All 7 Channels) | 16.3420 | 9.0937 | 0.0057 | 0.9035 | 2914.8764 |
| | Without FY-4A | 18.4221 | 10.7338 | −0.0131 | 0.8768 | 956.7337 |
| 20 min | FY-4A (Best 3 Channels) | 18.9787 | 11.7958 | −0.0651 | 0.8669 | 1894.0174 |
| | FY-4A (All 7 Channels) | 18.3483 | 11.110 | −0.0905 | 0.8756 | 2896.0017 |
| | Without FY-4A | 21.5889 | 14.2191 | −0.0153 | 0.8273 | 957.5262 |
| 30 min | FY-4A (Best 3 Channels) | 20.8671 | 13.7970 | −0.2842 | 0.8335 | 1703.3047 |
| | FY-4A (All 7 Channels) | 19.7538 | 12.6537 | −0.1426 | 0.8510 | 2604.9350 |
| | Without FY-4A | 23.8505 | 16.9326 | −0.1541 | 0.7821 | 1103.7431 |
| 40 min | FY-4A (Best 3 Channels) | 22.5741 | 15.3859 | −0.0080 | 0.7956 | 1957.5106 |
| | FY-4A (All 7 Channels) | 21.1893 | 14.0615 | 0.0619 | 0.8204 | 2860.9892 |
| | Without FY-4A | 26.2534 | 19.2950 | −0.0814 | 0.7244 | 995.3814 |
| 50 min | FY-4A (Best 3 Channels) | 24.1419 | 16.8739 | 0.0046 | 0.7625 | 1870.1118 |
| | FY-4A (All 7 Channels) | 22.5737 | 15.4720 | 0.0763 | 0.7934 | 2895.8723 |
| | Without FY-4A | 28.3193 | 21.4859 | 0.1587 | 0.6745 | 992.8946 |
| 60 min | FY-4A (Best 3 Channels) | 26.0161 | 18.5139 | −0.0345 | 0.7341 | 1757.2776 |
| | FY-4A (All 7 Channels) | 23.8498 | 16.6696 | 0.0262 | 0.7669 | 2733.2219 |
| | Without FY-4A | 29.8938 | 22.9646 | −0.0286 | 0.6166 | 959.7182 |
| 90 min | FY-4A (Best 3 Channels) | 29.9375 | 21.6341 | −0.7624 | 0.6100 | 1641.0330 |
| | FY-4A (All 7 Channels) | 27.1105 | 19.4680 | −0.5865 | 0.6813 | 2641.0437 |
| | Without FY-4A | 35.5776 | 28.0763 | −0.8342 | 0.4482 | 882.1576 |
| 120 min | FY-4A (Best 3 Channels) | 32.7330 | 23.7999 | −0.3325 | 0.5351 | 1521.5579 |
| | FY-4A (All 7 Channels) | 29.7598 | 21.2222 | −0.2448 | 0.6165 | 2517.6641 |
| | Without FY-4A | 39.6323 | 31.7468 | −0.3685 | 0.3163 | 872.8807 |
| 150 min | FY-4A (Best 3 Channels) | 34.0138 | 24.9394 | −0.3265 | 0.5048 | 1430.8113 |
| | FY-4A (All 7 Channels) | 30.5469 | 21.9532 | −0.4497 | 0.5994 | 2170.4864 |
| | Without FY-4A | 42.3607 | 34.5322 | −0.3514 | 0.2275 | 825.2293 |
| 180 min | FY-4A (Best 3 Channels) | 34.1602 | 25.2437 | −0.3288 | 0.5145 | 1266.1396 |
| | FY-4A (All 7 Channels) | 30.3670 | 21.9811 | −0.3159 | 0.6142 | 1985.8447 |
| | Without FY-4A | 44.1783 | 36.4969 | −0.7504 | 0.1795 | 740.1289 |

In summary, adding remote sensing data to input variables can significantly improve radiation predictions under different time steps. Full-channel input of remote sensing data can provide better prediction results, but if time and calculation cost are considered, the optimal three-channel model is a better choice.

### 3.4. Discussion

In addition to the method's accuracy, computational speed also plays a significant role in solar energy forecasts and, thus, should be considered here. The lightweight parameter input model computes the results more rapidly than the all-parameter input model. For instance, the lightweight parameter input model needed only 0.3084 s (compared to 0.5591 s for full-parameter input) per single data point on average for the 10 min global solar radiation forecast in Yuzhong when running on a computer with an i7-10700 CPU operating at 2.90 GHz and having 16 GB of RAM. As shown in Table 2, a similar conclusion can be drawn that models with lightweight input parameters had faster running speeds at different time horizons, and the time differences did not show a linear relationship with the time series. It is worth noting that the running speed of the model in the study is based on its implementation in Python language, while the running speed in other languages may be different.

It is worth noting that the running time of the models used in this study included the process of model establishment, super-parameter optimization and prediction. However, in the actual prediction process, the time proportion of model establishment and the super-parameter optimization were very high. Taking the 10 min radiation prediction in the Dunhuang area as an example, the running time of this part accounted for 99.9%. Therefore, if it is necessary to establish a prediction model in practical applications, the optimal channels (Channels 1, 2 and 7) can be selected for rapid radiation prediction. In contrast, if

there is an established radiation prediction model, more accurate predictions can be quickly with the full channel model.

## 4. Conclusions

Simple and accurate prediction methods are usually the first choice of users. In this regard, it is now clear that if the dependence on high spatial and temporal resolution observation data could be reduced as much as possible, this radiation prediction method might be used more widely. This paper addresses the following question: How can limited observation data be effectively used to obtain fast and accurate solar forecasting? To the best of our knowledge, there are few studies using radiation observations combined with satellite remote sensing data and McClear-based machine learning in northwestern China. Different from the previous research on the improvement of model and forecast effects, this study focuses on the influence of feature contributions on prediction effects and operation speeds.

A common problem in machine learning is how to allocate training and test data reasonably. To avoid the impact of sample differences on the prediction effect of the model, 70% of the data were randomly selected as the training data of the model by using the random module in Python. The results showed that it is reasonable and effective to divide the learning and prediction data in this way.

In addition, machine learning methods have once again proven to be an important means to provide reliable solar radiation prediction results. The results showed good behavior, especially regarding the general use of RF. For the overall prediction effect, the prediction effect of global solar radiation utilizing RF was better than that utilizing SVR. At the same time, according to the nMBE values, it can be clearly seen that using SVR to predict the value would have a greater probability of a positive deviation. For the radiation predictions in different time horizons, compared with the reference model (Clim-Pers) and SVR, RF performed the best, with its time horizon averages having the smallest nRMSE values (between 13.90% and 33.80%), smallest nMAE values (between 7.50% and 24.77%), smallest nMBE values (between $-1.17\%$ and 0.7%) and highest $R^2$ values (between 0.76 and 0.95).

More importantly, the study revealed the importance of using remote sensing data for adding radiation prediction results when only radiation observations are available. It is worth noting that the prediction effect without adding remote sensing data decreased rapidly after the time horizon exceeded 60 min. Based on the SHAP interpretation results, we selected three key channels (1, 2 and 7) on the FY-4A to improve the calculation speed of the model. The prediction results show that the differences in the predictions between the full-channel input and the three-channel input results did not change significantly with the time horizon. Therefore, on the premise of ensuring the computing ability, the full-channel input can provide more accurate prediction results. However, three-channel input can provide faster, accurate and reliable prediction results under the same conditions. The prediction time of 10 min per single data point on average in Yuzhong can be reduced from 0.5591 s to 0.3084 s.

**Author Contributions:** D.J.: conceptualization, writing—original draft, writing—review and editing, funding acquisition; L.Y.: data curation, software; X.G.: supervision, funding acquisition; K.L.: data curation. All authors have read and agreed to the published version of the manuscript.

**Funding:** This study was jointly supported by the Science and Technology Project of Gansu Province (No. 21JR7RA546) and Gansu Provincial Department of Education: University Teacher Innovation Fund Project (No. 2023B-151).

**Acknowledgments:** The authors would like to thank Chunxia Yuan, for the opportunity to work together with and for the help provided throughout this project.

**Conflicts of Interest:** The authors declare no conflict of interest.

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
