# Peer review of "Assessment of a New Solar Radiation Nowcasting Method Based on FY-4A Satellite Imagery, the McClear Model and SHapley Additive exPlanations (SHAP)"

_remotesensing, doi:10.3390/rs15092245_

Round 1

Reviewer 1 Report

This manuscript aims to develop a fast and accurate solar radiation calculation method by combining FY-4A satellite data and the McClear clear sky model along with machine learning techniques. The scientific objective is valuable, but how the authors achieve this goal is not clearly delivered in the current manuscript. This manuscript is very difficult to follow. There are many sections that need significant revision. I recommend rejection of this version and provide some comments and suggestions as listed below. The authors are encouraged to resubmit the manuscript after revision.

Major comments:

1.      Section 2.1: first, I don’t think this whole paragraph can be called research scope. Second, figure 1 is very difficult to see. Third, what are the predictors and responses in machine training for this study, and what is the reason for picking those predictors?

2.      Section 2.2.1: what is the time range of data used from FY-4A? Which seven channels are used in this study, and why those? Are geometries also used in this study? How do you calculate the albedo for each channel? What do you mean by missing cloud images?

3.      Section 2.2.2: need to describe more about the McClear data product, how you used this product in this study, and why you used it. What is the uncertainty associated with this product?

4.      Section 2.2.3: as a ground truth, I think at least the authors should introduce what kind of solar radiation data is measured from those ground stations and by which instruments. What is the uncertainty associated with the measured surface radiation? How do you compare this data against the machine learning prediction? Any preprocessing of this data before you can make a comparison? Ground observation of radiation can be easily affected by the scattered clouds and shades, which is a function of cloud fraction, cloud position, and solar zenith angle. How do you account for these effects when comparing against the models?

5.      Section 2.3.1: the structure of this section seems weird. I appreciate the authors providing as many details as possible regarding machine learning algorithms. However, the descriptions in this section seem too broad. What makes the two machine learning algorithms unique to be picked for this application?

6.      Section 2.3: after reading through this section, it’s unclear how the authors design the framework for forecasting solar radiation from satellite data and machine learning algorithms.

7.      Section 3: Figures 3, 4, 7, and 8 are blurry and very difficult to see.

8.      English writing needs revision. 

Author Response

Major comments:

  1. Section 2.1: first, I don’t think this whole paragraph can be called research scope. Second, figure 1 is very difficult to see. Third, what are the predictors and responses in machine training for this study, and what is the reason for picking those predictors?

Responds: Thanks for carefully reviewing this manuscript. According to your suggestion, we have changed the title of section 2.1 to "Introduction of the research process". In addition, all images in the manuscript have been replaced with vector images to increase their clarity. We’re sorry for the inconvenience caused by the unclear description. For machine learning, our original intention is to use remote sensing data to supplement the shortcomings of only radiation observation data. Therefore, we use radiation observation data, remote sensing observation data (different channel combinations), and McClear data (used as clear sky radiation correction) as input data, and use radiation observation data at different time horizon as output variables. Considering your suggestion, we have added a description of machine learning input and output parameters to the manuscript, which is ‘The radiation observation data at the current time, remote sensing data (different band combinations), and McClear data are used as input parameters for machine learning, and the radiation observation data corresponding to the predicted time step is used as output parameters’. All additions have been marked red in the manuscript.

  1. Section 2.2.1: what is the time range of data used from FY-4A? Which seven channels are used in this study, and why those? Are geometries also used in this study? How do you calculate the albedo for each channel? What do you mean by missing cloud images?

Responds: Thanks for your questions. The time interval of the image data of FY4A is not completely equal, which between 10 minutes and 25 minutes. Therefore, we process the remote sensing data as follows: First, we linearly interpolate the data from different pixel points to obtain the average albedo region at a time resolution of 10 minutes for each of the three channels. The cloud image clipping area is 32 km × At 32 km, the pixel points in the inversion area are centered, and the missing cloud image is interpolated using a linear interpolation method. Finally, the regional average albedo values of seven channels with a time resolution of 10 minutes are obtained.

  Based on your suggestion, we have added the following content to section 2.2.1 and marked it in red, which is ‘FY-4A carries a variety of payloads, including an advanced geosynchronous radiation imager (AGRI; 14 channels between 0.47 and 13.5 µm, with a spatial resolution of 0.5–4 km), an interferometric atmospheric vertical detector (GIIRS), a lightning imager (LMI), and a space environment monitoring instrument (SEP). The spatial resolution of satellite images used in this study is 2 km×2 km, and seven channels (0.45 ~ 4.00 μ m; three channels for visible light, three channels for short wave infrared, and one channel for medium wave infrared) are selected’. All changes have been marked red in the manuscript.

  1. Section 2.2.2: need to describe more about the McClear data product, how you used this product in this study, and why you used it. What is the uncertainty associated with this product?

Responds: Thanks for carefully reviewing the manuscript. The purpose of using McClear in the manuscript is to provide fast and accurate clear sky radiation data for the machine learning input. The global, direct, and diffuse horizontal irradiation, as well as the beam normal irradiation of 1min, 15min, 1 hour, daily and monthly time scales can be obtained directly by inputting longitude, latitude, altitude and output format (https://www.soda-pro.com/web-services/radiation/cams-mcclear). Based on your suggestion, we have added a more detailed description of McClear data in section 2.2.2, which is ‘McClear is a is a fully physical model parameterized by A, z, several parameters describing the optical state of the atmosphere. For clear-sky conditions, an irradiation time series is provided for any location in the world using information on aerosol, ozone and water vapour from the CAMS global forecasting system. Other properties, such as ground albedo and ground elevation, are also considered. Similar time series are available for cloudy conditions but, since the high-resolution cloud information is directly inferred from satellite observations, these are currently only available inside the field-of-view of the Meteosat Second Generation (MSG) satellite, which is roughly Europe, Africa, the Atlantic Ocean and the Middle East. Input quality control, regular quarterly benchmarking against ground stations, and regular monitoring the consistency and detecting possible trends is performed’. All additions have been marked red in the manuscript.

  1. Section 2.2.3: as a ground truth, I think at least the authors should introduce what kind of solar radiation data is measured from those ground stations and by which instruments. What is the uncertainty associated with the measured surface radiation? How do you compare this data against the machine learning prediction? Any preprocessing of this data before you can make a comparison? Ground observation of radiation can be easily affected by the scattered clouds and shades, which is a function of cloud fraction, cloud position, and solar zenith angle. How do you account for these effects when comparing against the models?

Responds: Thanks for your questions. According to your suggestions, we have added the related instruments details, which is ‘According to the standards of ISO 9060:1990 and WMO, all instruments are calibrated and maintained in time. The dataset used was from July 2019 to June 2020 and the temporal resolution was 1 min. These stations provided data of short-wave downward global radiation in W/m2 which measured using the FS-S6 solar radiation sensor. The observation data in input data are real-time matched with remote sensing data, while the radiation data in output data are delayed matched based on the predicted time step’.

Radiation observation data are used in machine model applications as part of the input data and as output validation data. Therefore, the solar radiation which affected by natural factors such as clouds, we have also input the radiation observation data, clear sky data, and satellite data into the machine learning model to establish a relationship. The specific operation process is described below. First, the data has undergone quality control and pretreatment. The radiation observation data at the current time, remote sensing data (different band combinations), and McClear data are used as input parameters for machine learning, and the radiation observation data corresponding to the predicted time step is used as output parameters.

  1. Section 2.3.1: the structure of this section seems weird. I appreciate the authors providing as many details as possible regarding machine learning algorithms. However, the descriptions in this section seem too broad. What makes the two machine learning algorithms unique to be picked for this application?

Responds: Thanks for carefully reviewing the manuscript. Section 2.3.1 mainly describes the core algorithms in these two machine learning models. Based on your suggestion, after discussion, we have added relevant content at the beginning of section 2.3.1, which is ‘According to previous research [3], SVR and RF are the best two models for estimating and predicting solar radiation in China region. Therefore, this study used the following two methods to construct radiation prediction models, and then selected the optimal model from them’.

  1. Section 2.3: after reading through this section, it’s unclear how the authors design the framework for forecasting solar radiation from satellite data and machine learning algorithms.

Responds: Thanks for your questions. Section 2.3 is mainly used to introduce the key algorithms within two machine models. The specific design and operational details are described in detail in section 2.1, which is ‘The operation steps of the method are briefly shown in in Fig. 1. First, the data has undergone quality control and pretreatment. The radiation observation data at the current time, remote sensing data (different band combinations), and McClear data are used as input parameters for machine learning, and the radiation observation data corresponding to the predicted time step is used as output parameters. Data were randomly grouped into training and testing datasets at proportions of 70% and 30%, respectively. Next, the RF and SVR algorithms were introduced for model building. To find the optimal parameters, the grid search method was adopted to derive the hyperparameters. Furthermore, performance evaluations were performed based on statistical indicators. Related indicators, such as nMAE, nMBE, nRMSE, Skillscore and R, were utilized to quantify the evaluation results. Then, the optimal machine learning method was selected based on the comprehensive performance. Finally, in the interpretation and lightweight process, the lightweight parameter input model (based on the SHAP results) was established, and a comparison with the initial model in the same field was also made’. The modifications and additions have been highlighted in red in the manuscript.

  1. Section 3: Figures 3, 4, 7, and 8 are blurry and very difficult to see.

Responds: Thanks for your suggestions. According to your suggestion, Figures 3 and 4 have been replaced with vector diagrams, and Figures 7 and 8 have been combined and redrawn

  1. English writing needs revision. 

Responds: Thanks for your suggestions. The manuscript has previously undergone language editing services, but there may be language irregularities due to content additions and modifications. We have contacted native speaker to modify the manuscript again. I hope the modified content can meet the requirements.

Reviewer 2 Report

In manuscript a fast and accurate solar radiation nowcasting method is proposed by combining FY-4A satellite data and the McClear clear sky model under the condition of only radiation observation. It is shown that remote sensing data can significantly improve the radiation forecasting performance and can effectively guarantee the stability of radiation prediction when the time horizon exceeds 60 min. According authors, radiation prediction is very important in sites with scarce meteorological observation for choosing the location of  planning photovoltaic power stations. I thing that meteorological observation is fully sufficient for choosing the location of photovoltaic power stations and using sattelite is not reasonable.

It will be noted that Figures 1,2,4,7,8 are of poor quality, it is almost impossible to read the notes.

The manuscript can be accepted after amendment.

Author Response

In manuscript a fast and accurate solar radiation nowcasting method is proposed by combining FY-4A satellite data and the McClear clear sky model under the condition of only radiation observation. It is shown that remote sensing data can significantly improve the radiation forecasting performance and can effectively guarantee the stability of radiation prediction when the time horizon exceeds 60 min. According authors, radiation prediction is very important in sites with scarce meteorological observation for choosing the location of  planning photovoltaic power stations. I thing that meteorological observation is fully sufficient for choosing the location of photovoltaic power stations and using sattelite is not reasonable.

It will be noted that Figures 1,2,4,7,8 are of poor quality, it is almost impossible to read the notes.

The manuscript can be accepted after amendment.

Responds: Thanks for carefully reviewing this manuscript, and your suggestions are very important for improving this article. According to previous research results, meteorological data (including water vapor, temperature and so on) can be used as input parameters together with radiation observations, which will improve the effectiveness of radiation prediction. Therefore, the assumption of this study is how to provide accurate and reliable radiation prediction data, with only radiation observations without meteorological data. Due to the sparse observation stations in western China, this will greatly reduce the early observation costs. Radiation observation data is one of the essential key parameters for radiation prediction. However, remote sensing data can optimize radiation prediction effects and provide more stable prediction results. Table 2 shows that adding satellite data to the input data can significantly improve the radiation predictions. It is worth noting that the prediction effect without adding remote sensing data decreased rapidly after the time horizon exceeded 60 min. The specific details are that the values of nRMSE and nMAE continued to increase, and the prediction accuracy decreased rapidly. Among them, the range of nRMSE differences (excluding remote sensing data minus including remote sensing data) changed from 3.8777%~6.044% with a time horizon of 60 min to 10.0181%~13.8113% with a time horizon of 180 min; R2 changed from 0.6166 with a time horizon of 60 min to 0.1795 with a time horizon of 180 min. Therefore, it can be concluded that remote sensing data can improve the radiation predictions and effectively guarantee the stability of radiation prediction when the time horizon exceeds 60 min.

  In addition, according to your suggestion, all images in the manuscript have been replaced with vector images to increase their clarity.

Reviewer 3 Report

This manuscript proposed a random forest-based method to nowcast solar radiation at three ground stations from geostationary satellite data. Considering the difficulty in obtaining meteorological ground observations at large spatial scales, this study shows the potential that only utilizing limited satellite observations can also achieve higher prediction accuracy. This work is interesting; however, the figure qualities are low, making it hard to be reviewed. The literature review and method description should be also revised. I would recommend the authors reorganize the manuscript and submit it again.

1. Literature review

The authors divided the previous methodologies into three categories based on the input data. Based on the critical summary, they claimed the drawbacks of the satellite data-based studies "are limited in time and space due to their grid size and output time". This is correct but not enough because such weakness is inevitable in satellite data-based studies and this study also suffers from it. I would suggest adding one more paragraph to summarize all satellite data-based literature specifically, and point out their drawbacks and what makes you different from them.

2. Data

The input data introduction missed some key information. As this study is essential to predict solar radiation thus it is necessary to provide readers with the input data latency in real-time. The band spectral information of FY-4A is also important. 

The McClear Data should be introduced in detail. How is it generated? What is the data source? spatiotemporal resolution? any latency?

3. Result section

I don't if it is the internet issue I cannot see the figures clearly at all in the PDF file. Please revise the manuscript then I can continue reviewing it.

All figures should mark (a) (b) (c) for subplots and point out the meanings of the subplots in captions. There are two 3.1 sections in this section and the first one seems not relevant to the test for different time horizons.

Accuracies should be compared with previous ML-based solar radiation prediction analyses.

4. Abstract

The abstract section should include some quantitative results to demonstrate the accuracy of the proposed model. 

Author Response

  1. Literature review

The authors divided the previous methodologies into three categories based on the input data. Based on the critical summary, they claimed the drawbacks of the satellite data-based studies "are limited in time and space due to their grid size and output time". This is correct but not enough because such weakness is inevitable in satellite data-based studies and this study also suffers from it. I would suggest adding one more paragraph to summarize all satellite data-based literature specifically, and point out their drawbacks and what makes you different from them.

Responds: Thanks for your suggestions. The differences between this study and the previous use of remote sensing radiation have been explained in the last paragraph of the introduction. Therefore, a summary and review of relevant literature have been added here to explain the existing issues and future development trends in radiation research using remote sensing data. The additional parts are as follows, which is ‘In terms of estimation algorithms, the existing universally applicable methods is relatively high on a large scale, but their calculation speed is greatly reduced when the number of input parameters is large. Consequently, current methods cannot easily meet the needs of real-time observation and cannot cope with massive satellite data. In addition, under the premise of ensuring inversion accuracy, attention should be given to the efficient of data and improvement of inversion efficiency in the future, such as through the efficient combination of look-up tables (LUTs) methods and deep learning, and other artificial intelligence method to promote the development of high-precision solar radiation products [11-14]’. All additions have been marked red in the manuscript.

  1. Data

The input data introduction missed some key information. As this study is essential to predict solar radiation thus it is necessary to provide readers with the input data latency in real-time. The band spectral information of FY-4A is also important. 

The McClear Data should be introduced in detail. How is it generated? What is the data source? spatiotemporal resolution? any latency?

Responds: Thanks for your suggestions, which are important for improving the manuscript. According to your suggestion, we have added the following sentence to the section 2.2.3 Observation part, which is “The observation data in input data are real-time matched with remote sensing data, while the radiation data in output data are delayed matched based on the predicted time step”. More detailed information on FY-4A and McClear have also been added, which is ‘FY-4A carries a variety of payloads, including an advanced geosynchronous radiation imager (AGRI; 14 channels between 0.47 and 13.5 µm, with a spatial resolution of 0.5–4 km), an interferometric atmospheric vertical detector (GIIRS), a lightning imager (LMI), and a space environment monitoring instrument (SEP)’ and ‘McClear is a is a fully physical model parameterized by A, z, several parameters describing the optical state of the atmosphere. For clear-sky conditions, an irradiation time series is provided for any location in the world using information on aerosol, ozone and water vapour from the CAMS global forecasting system. Other properties, such as ground albedo and ground elevation, are also considered. Similar time series are available for cloudy conditions but, since the high-resolution cloud information is directly inferred from satellite observations, these are currently only available inside the field-of-view of the Meteosat Second Generation (MSG) satellite, which is roughly Europe, Africa, the Atlantic Ocean and the Middle East. Input quality control, regular quarterly benchmarking against ground stations, and regular monitoring the consistency and detecting possible trends is performed’. All additions have been marked red in the manuscript.

  1. Result section

I don't if it is the internet issue I cannot see the figures clearly at all in the PDF file. Please revise the manuscript then I can continue reviewing it.

All figures should mark (a) (b) (c) for subplots and point out the meanings of the subplots in captions. There are two 3.1 sections in this section and the first one seems not relevant to the test for different time horizons.

Accuracies should be compared with previous ML-based solar radiation prediction analyses.

Responds: Thanks for your suggestions, which are important for improving the manuscript. Based on your suggestion, we have revised the duplicate section titles and added content related to the comparison of predicted results in the manuscript, which is ‘Compared with the research results of Ravinesh et al. [27], both methods can provide radiation prediction within an acceptable range. However, our method can provide more stable radiation prediction results as the time step increases, and requires fewer operational and input parameters’. Meanwhile, the first paragraph of section 3.1 discusses how to effectively utilize random functions to allocate pattern learning and test data in pattern construction. Based on your suggestion, we have moved this section to before section 3.1. All reference additions and corrections in the text have been highlighted in red.

  The addition of subplots can indeed improve the readability of images, but referring to the group images in journals such as Solar Energy and Journal of Renewable and Sustainable Energy, it was found that when the vertical and subheadings can clearly explain the meaning of the image, there is no need to add subplots. Therefore, we did not modify the group images in the manuscript when the subheadings and vertical titles can clearly reflect the meaning of the image.

  1. Abstract

The abstract section should include some quantitative results to demonstrate the accuracy of the proposed model. 

Responds: Thanks for carefully reviewing the manuscript. After discussion, we have added some quantitative results to demonstrate the accuracy and operation speed in the abstract part, which is ‘The results show that random forest (RF) performs better than the support vector regression (SVR) model and the reference model (Clim-pers), with the smallest nRMSE values (between 13.90% and 33.80%), smallest nMAE values (between 7.50% and 24.77%), smallest nMBE values (between -1.17% and 0.7%), and highest R2 values (between 0.76 and 0.95) under different time horizon’ and ‘For instance, the lightweight parameter input model needed only 0.3084 s (compared to 0.5591 s for full parameter input) per single data point on average for the 10 min global solar radiation forecast in Yuzhong’. All modifications are marked red in the manuscript.

Round 2

Reviewer 1 Report

I would suggest revising the abstract to use "normalized RMSE" instead of "nRMSE," as well as "normalized" instead of "normalised" throughout the manuscript. The same should be done for MBE and MAE. With these minor corrections, I recommend accepting this manuscript.

Author Response

Responds: Thanks for carefully reviewing this manuscript. We believe that your suggestions are crucial for improving the quality of this manuscript. After discussion, the nMAE, nRMSE, and nMBE in our abstract have been revised to their full names and abbreviations, respectively, which is ‘The results show that random forest (RF) performs better than the support vector regression (SVR) model and the reference model (Clim-pers), with the smallest normalized root mean square error (nRMSE) values (between 13.90% and 33.80%), smallest normalized mean absolute error (nMAE) values (between 7.50% and 24.77%), smallest normalized mean bias error (nMBE) values (between -1.17% and 0.7%), and highest R2 values (between 0.76 and 0.95) under different time horizon’. Meanwhile, we have revised the word 'normalised' to 'normalized' in the manuscript. All the changes have been marked red in the manuscript.

Reviewer 2 Report

The revised manuscript has the same flaws in presentation that I pointed out earlier.

Figures 1,2,4,8 are of poor quality, it is almost impossible to read the notes. In addition, Fig. 7 is canceled so Fig. 8 should be marked with number 7.

The manuscript can be accepted after amendment.

Author Response

Responds: Thanks for carefully reviewing this manuscript. Based on your suggestion, we have improved the image quality. Figures 1, 2, and 4 have all been redrawn and the font size has been adjusted to allow readers to read the image more clearly. Due to the merging of Figure 7 and Figure 8, we have modified the image title and adjusted the font size. We hope that the modifications can meet your requirements. If there are any issues with the image in the future, we will actively contact the editor to make corrections.

Reviewer 3 Report

The manuscript has been improved but the image quality is still low, please work with assistant editors to revise this issue.

Author Response

(The authors gave the same response as above.)
